# Volumetric Convolution: Automatic Representation Learning in Unit Ball

## Abstract

Convolution is an efficient technique to obtain abstract feature representations using hierarchical layers in deep networks. Although performing convolution in Euclidean geometries is fairly straightforward, its extension to other topological spaces—such as a sphere ($\mathbb{S}^2$) or a unit ball ($\mathbb{B}^3$)—entails unique challenges. In this work, we propose a novel '*volumetric convolution*' operation that can effectively convolve arbitrary functions in $\mathbb{B}^3$. We develop a theoretical framework for *volumetric convolution* based on Zernike polynomials and efficiently implement it as a differentiable and an easily pluggable layer for deep networks. Furthermore, our formulation leads to derivation of a novel formula to measure the symmetry of a function in $\mathbb{B}^3$ around an arbitrary axis, that is useful in 3D shape analysis tasks. We demonstrate the efficacy of proposed volumetric convolution operation on a possible use-case i.e., 3D object recognition task.

## 1 Introduction

Convolution-based deep neural networks have performed exceedingly well on 2D representation learning tasks (Krizhevsky et al., 2012; He et al., 2016). The convolution layers perform parameter sharing to learn repetitive features across the spatial domain while having lower computational cost by using local neuron connectivity. However, most state-of-the-art convolutional networks can only work on Euclidean geometries and their extension to other topological spaces e.g., spheres, is an open research problem. Remarkably, the adaptation of convolutional networks to spherical domain can advance key application areas such as robotics, geoscience and medical imaging.

Some recent efforts have been reported in the literature that aim to extend convolutional networks to spherical signals. Initial progress was made by Boomsma & Frellsen (2017), who performed conventional planar convolution with a careful padding on a spherical-polar representation and its cube-sphere transformation (Ronchi et al., 1996). A recent pioneering contribution by Cohen et al. (2018) used harmonic analysis to perform efficient convolution on the surface of the sphere ($\mathbb{S}^2$) to achieve rotational equivariance. These works, however, do not systematically consider radial information in a 3D shape and the feature representations are learned at specified radii. Specifically, Cohen et al. (2018) estimated similarity between spherical surface and convolutional filter in $\mathbb{S}^2$, where the kernel can be translated in $\mathbb{S}^2$. Furthermore, Weiler et al. (2018) recently solved the more general problem of SE(3) equivariance by modeling 3D data as dense vector fields in 3D Euclidean space. In this work however, we focus on $\mathbb{B}^3$ to achieve the equivariance to SO(3).

In this paper, we propose a novel approach to perform *volumetric convolutions* inside unit ball ($\mathbb{B}^3$) that explicitly learns representations across the radial axis. Although we derive generic formulas to convolve functions in $\mathbb{B}^3$, we experiment on one possible use case in this work, i.e., 3D shape recognition. In comparison to closely related spherical convolution approaches, modeling and convolving 3D shapes in $\mathbb{B}^3$ entails two key advantages: '*volumetric convolution*' can capture both 2D texture and 3D shape features and can handle non-polar 3D shapes. We develop the theory of volumetric convolution using orthogonal Zernike polynomials (Canterakis, 1999), and use careful approximations to efficiently implement it using low computational-cost matrix multiplications. Our experimental results demonstrate significant boost over spherical convolution and that confirm the high discriminative ability of features learned through volumetric convolution.

Furthermore, we derive an explicit formula based on Zernike Polynomials to measure the axial symmetry of a function in $\mathbb{B}^3$, around an arbitrary axis. While this formula can be useful in many

function analysis tasks, here we demonstrate one particular use-case with relevance to 3D shape recognition. Specifically, we use the the derived formula to propose a hand-crafted descriptor that accurately encodes the axial symmetry of a 3D shape. Moreover, we decompose the implementation of both volumetric convolution and axial symmetry measurement into differentiable steps, which enables them to be integrated to any end-to-end architecture.

Finally, we propose an experimental architecture to demonstrate the practical usefulness of proposed operations. We use a capsule network after the convolution layer as it allows us to directly compare feature discriminability of spherical convolution and volumetric convolution without any bias. In other words, the optimum deep architecture for spherical convolution may not be the same for volumetric convolution. Capsules, however, do not deteriorate extracted features and the final accuracy only depends on the richness of input shape features. Therefore, a fair comparison between spherical and volumetric convolutions can be done by simply replacing the convolution layer.

It is worth pointing out that the proposed experimental architecture is only a one possible example out of many possible architectures, and is primarily focused on three factors: *1)* Capture useful features with a relatively shallow network compared to state-of-the-art. *2)* Show richness of computed features through clear improvements over spherical convolution. *3)* Demonstrate the usefulness of the volumetric convolution and axial symmetry feature layers as fully differentiable and easily pluggable layers, which can be used as building blocks for end-to-end deep architectures.

The main contributions of this work include:

- Development of the theory for volumetric convolution that can efficiently model functions in $\mathbb{B}^3$.
- Implementation of the proposed volumetric convolution as a fully differentiable module that can be plugged into any end-to-end deep learning framework.
- The first approach to perform volumetric convolution on 3D objects that can simultaneously model 2D (appearance) and 3D (shape) features.
- A novel formula to measure the axial symmetry of a function defined in $\mathbb{B}^3$, around an arbitrary axis using Zernike polynomials.
- An experimental end-to-end trainable framework that combines hand-crafted feature representation with automatically learned representations to obtain rich 3D shape descriptors.

The rest of the paper is structured as follows. In Sec. 2 we introduce the overall problem and our proposed solution. Sec. 3 presents an overview of 3D Zernike polynomials. Then, in Sec. 4 and Sec. 5 we derive the proposed volumetric convolution and axial symmetry measurement formula respectively. Sec. 6.2 presents our experimental architecture, and in Sec. 7 we show the effectiveness of the derived operators through extensive experiments. Finally, we conclude the paper in Sec. 8.

## 2 PROBLEM DEFINITION

Convolution is an effective method to capture useful features from uniformly spaced grids in $\mathbb{R}^n$, within each dimension of $n$, such as gray scale images ($\mathbb{R}^2$), RGB images ($\mathbb{R}^3$), spatio-temporal data ($\mathbb{R}^3$) and stacked planar feature maps ($\mathbb{R}^n$). In such cases, uniformity of the grid within each dimension ensures the translation equivariance of the convolution. However, for topological spaces such as $\mathbb{S}^2$ and $\mathbb{B}^3$, it is not possible to construct such a grid due to non-linearity. A naive approach to perform convolution in $\mathbb{B}^3$ would be to create a uniformly spaced three dimensional grid in $(r, \theta, \phi)$ coordinates (with necessary padding) and perform 3D convolution. However, the spaces between adjacent points in each axis are dependant on their absolute position and hence, modeling such a space as a uniformly spaced grid is not accurate.

To overcome these limitations, we propose a novel *volumetric convolution* operation which can effectively perform convolution on functions in $\mathbb{B}^3$. It is important to note that ideally, the convolution in $\mathbb{B}^3$ should be a signal on both 3D rotation group and 3D translation. However, since Zernike polynomials do not have the necessary properties to automatically achieve translation equivariance, we stick to 3D rotation group in this work and refer to this operation as convolution from here onwards. Fig. 1 shows the analogy between planar convolution and volumetric convolution. In Sec. 3, we present an overview of 3D Zernike polynomials that will be later used in Sec. 4 to develop volumetric convolution operator.

## 3  3D ZERNIKE POLYNOMIALS

3D Zernike polynomials are a complete and orthogonal set of basis functions in $\mathbb{B}^3$, that exhibits a '*form invariance*' property under 3D rotation (Canterakis, 1999). A $(n, l, m)^{th}$ order 3D Zernike basis function is defined as,

$$Z_{n,l,m} = R_{n,l}(r)Y_{l,m}(\theta, \phi) \tag{1}$$

where $R_{n,l}$ is the Zernike radial polynomial (Appendix D.3), $Y_{l,m}(\theta, \phi)$ is the spherical harmonics function (Appendix D.1), $n \in \mathbb{Z}^+$, $l \in [0, n]$, $m \in [-l, l]$ and $n - l$ is even. Since 3D Zernike polynomials are orthogonal and complete in $\mathbb{B}^3$, an arbitrary function $f(r, \theta, \phi)$ in $\mathbb{B}^3$ can be approximated using Zernike polynomials as follows.

$$f(\theta, \phi, r) = \sum_{n=0}^{\infty} \sum_{l=0}^{n} \sum_{m=-l}^{l} \Omega_{n,l,m}(f)Z_{n,l,m}(\theta, \phi, r) \tag{2}$$

where $\Omega_{n,l,m}(f)$ could be obtained using,

$$\Omega_{n,l,m}(f) = \int_0^1 \int_0^{2\pi} \int_0^{\pi} f(\theta, \phi, r)Z_{n,l,m}^{\dagger} r^2 sin\phi dr d\phi d\theta \tag{3}$$

where $\dagger$ denotes the complex conjugate. In Sec. 4, we will derive the proposed volumetric convolution.

## 4  VOLUMETRIC CONVOLUTION OF FUNCTIONS IN $\mathbb{B}^3$

When performing convolution in $\mathbb{B}^3$, a critical problem which arises is that several rotation operations exist for mapping a point $p$ to a particular point $p'$. For example, using Euler angles, we can decompose a rotation into three rotation operations $R(\theta, \phi) = R(\theta)_y R(\phi)_z R(\theta)_y$, and the first rotation $R(\theta)_y$ can differ while mapping $p$ to $p'$ (if $y$ is the north pole). However, if we enforce the kernel function to be symmetric around $y$, the function of the kernel after rotation would only depend on $p$ and $p'$. This observation is important for our next derivations because we can then uniquely define a 3D rotation on kernel in terms of azimuth and polar angles.

Let the kernel be symmetric around $y$ and $f(\theta, \phi, r)$, $g(\theta, \phi, r)$ be the functions of object and kernel respectively. Then we define volumetric convolution as,

$$f * g(\alpha, \beta) := \langle f(\theta, \phi, r), \tau_{(\alpha, \beta)}(g(\theta, \phi, r)) \rangle = \int_0^1 \int_0^{2\pi} \int_0^{\pi} f(\theta, \phi, r), \tau_{(\alpha, \beta)}(g(\theta, \phi, r)) \sin \phi d\phi d\theta dr \tag{4}$$

where $\tau_{(\alpha, \beta)}$ is an arbitrary rotation, that aligns the north pole with the axis towards $(\alpha, \beta)$ direction ($\alpha$ and $\beta$ are azimuth and polar angles respectively). Eq. 4 is able to capture more complex patterns compared to spherical convolution due to two reasons: *1)* the inner product integrates along the radius and *2)* the projection onto spherical harmonics forces the function into a polar function, that can result in information loss.

In Sec. 4.1 we derive differentiable relations to compute 3D Zernike moments for functions in $\mathbb{B}^3$.

### 4.1  SHAPE MODELING OF FUNCTIONS IN $\mathbb{B}^3$ USING 3D ZERNIKE POLYNOMIALS

Instead of using Eq. 3, we derive an alternative method to obtain the set $\{\Omega_{n,l,m}\}$. The motivations are two fold: *1)* ease of computation and *2)* the completeness property of 3D Zernike Polynomials ensures that $\lim_{n \to \infty} \|f - \sum_n \sum_l \sum_m \Omega_{n,l,m} Z_{n,l,m}\| = 0$ for any arbitrary function $f$. However, since $n$ should be finite in the implementation, aforementioned property may not hold, leading to increased distance between the Zernike representation and the original shape. Therefore, minimizing the reconstruction error $\sum_{(\theta, \phi, r) \in \mathbb{S}^3} |\bar{f}(\theta, \phi, r) - f(\theta, \phi, r)|$ where $\bar{f}(\theta, \phi, r) = \sum_n^N \sum_l \sum_m \Omega_{n,l,m} Z_{n,l,m}$, pushes the set $\{\Omega_{n,l,m}\}$ inside frequency space, where $\{\Omega_{n,l,m}\}$ has a closer resemblance to the corresponding shape. Following this conclusion, we derive the following method to obtain $\{\Omega_{n,l,m}\}$.

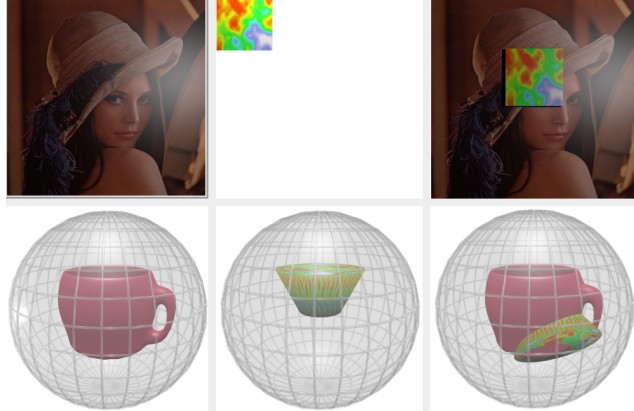

Figure 1: Analogy between planar and volumetric convolutions. Top *(left to right)*: image, kernel and planar convolution. Bottom *(left to right)*: 3D object, 3D kernel and volumetric convolution. In planar convolution the kernel translates and inner product between the image and the kernel is computed in $(x, y)$ plane. In volumetric convolution a 3D rotation is applied to the kernel and the inner product is computed between 3D function and 3D kernel over $\mathbb{B}^3$.

Since $Y_{l,m}(\theta, \phi) = (-1)^m \sqrt{\frac{2l+1}{4\pi} \frac{(l-m)!}{(l+m)!}} P_l^m(cos\phi)e^{im\theta}$, where $P_l^m(\cdot)$ is the associated Legendre function (Appendix D.2), it can be deduced that, $Y_{l,-m}(\theta, \phi) = (-1)^m Y_{l,m}^\dagger(\theta, \phi)$. Using this relationship we obtain $Z_{n,l,-m}(\theta, \phi) = (-1)^m Z_{n,l,m}^\dagger(\theta, \phi)$ and hence approximate Eq. 2 as,

$$f(\theta, \phi, r) = \sum_{n=0}^{\infty} \sum_{l=0}^{n} \sum_{m=0}^{l} A_{n,l,m} Re\{Z_{n,l,m}\} + B_{n,l,m} Img\{Z_{n,l,m}\} \tag{5}$$

where $Re\{Z_{n,l,m}\}$ and $Img\{Z_{n,l,m}\}$ are real and imaginary components of $Z_{n,l,m}$ respectively. In matrix form, this can be rewritten as,

$$f(\theta, \phi, r) = Ua + Vb = f(\theta, \phi, r) = Xc \tag{6}$$

where $c$ is the set of 3D Zernike moments $\Omega_{n,l,m}$. Eq. 6 can be interpreted as an overdetermined linear system, with the set $\Omega_{n,l,m}$ as the solution. To find the least squared error solution to the Eq. 6 we use the pseudo inverse of $X$. Since this operation has to be differentiable to train the model end-to-end, a common approach like singular value decomposition cannot be used here. Instead, we use an iterative method to calculate the pseudo inverse of a matrix (Li et al., 2011). It has been shown that $V_n$ converges to $A^+$ where $A^+$ is the Moore-Penrose pseudo inverse of $A$ if,

$$V_{n+1} = V_n(3I - AV_n(3I - AV_n)), n \in \mathbb{Z}^+ \tag{7}$$

for a suitable initial approximation $V_0$. They also showed that a suitable initial approximation would be $V_0 = \alpha A^T$ with $0 < \alpha < 2/\rho(AA^T)$, where $\rho(\cdot)$ denotes the spectral radius. Empirically, we choose $\alpha = 0.001$ in our experiments. Next, we derive the theory of volumetric convolution within the unit ball.

## 4.2 CONVOLUTION IN $\mathbb{B}^3$ USING 3D ZERNIKE POLYNOMIALS

We formally present our derivation of volumetric convolution using the following theorem. A short version of the proof is then provided. Please see Appendix A for the complete derivation.

**Theorem 1:** *Suppose $f, g : X \longrightarrow \mathbb{R}^3$ are square integrable complex functions defined in $\mathbb{B}^3$ so that $\langle f, f \rangle < \infty$ and $\langle g, g \rangle < \infty$. Further, suppose $g$ is symmetric around north pole and $\tau(\alpha, \beta) = R_y(\alpha)R_z(\beta)$ where $R \in SO(3)$. Then,*

$$\int_0^1 \int_0^{2\pi} \int_0^{\pi} f(\theta, \phi, r), \tau_{(\alpha,\beta)}(g(\theta, \phi, r)) \sin\phi d\phi d\theta dr \equiv \frac{4\pi}{3} \sum_{n=0}^{\infty} \sum_{l=0}^{n} \sum_{m=-l}^{l} \Omega_{n,l,m}(f)\Omega_{n,l,0}(g)Y_{l,m}(\theta, \phi)$$

$$\tag{8}$$

*where $\Omega_{n,l,m}(f), \Omega_{n,l,0}(g)$ and $Y_{l,m}(\theta, \phi)$ are $(n, l, m)^{th}$ 3D Zernike moment of $f$, $(n, l, 0)^{th}$ 3D Zernike moment of $g$, and spherical harmonics function respectively.*

**Proof:** Completeness property of 3D Zernike Polynomials ensures that it can approximate an arbitrary function in $\mathbb{B}^3$, as shown in Eq. 2. Leveraging this property, Eq. 4 can be rewritten as,

$$f * g(\theta, \phi) = \langle \sum_{n=0}^{\infty} \sum_{l=0}^{n} \sum_{m=-l}^{l} \Omega_{n,l,m}(f)Z_{n,l,m}, \tau_{(\theta,\phi)}(\sum_{n'=0}^{\infty} \sum_{l'=0}^{n'} \sum_{m'=-l}^{l} \Omega_{n',l',m'}(g)Z_{n',l',m'}) \rangle \tag{9}$$

However, since $g(\theta, \phi, r)$ is symmetric around $y$, the rotation around $y$ should not change the function. This ensures,

$$g(r, \theta, \phi) = g(r, \theta - \alpha, \phi) \tag{10}$$

and hence,

$$\sum_{n'=0}^{\infty} \sum_{l'=0}^{n'} \sum_{m'=-l}^{l} \Omega_{n',l',m'}(g) R_{n',l'}(r) Y_{l',m'} = \sum_{n'=0}^{\infty} \sum_{l'=0}^{n'} \sum_{m'=-l}^{l} \Omega_{n',l',m'}(g) R_{n',l'}(r) Y_{l',m'} e^{-im'\alpha} \tag{11}$$

This is true, if and only if $m' = 0$. Therefore, a symmetric function around $y$, defined inside the unit sphere can be rewritten as,

$$\sum_{n'=0}^{\infty} \sum_{l'=0}^{n'} \Omega_{n',l',0}(g) Z_{n',l',0} \tag{12}$$

which simplifies Eq. 9 to,

$$f * g(\theta, \phi) = \langle \sum_{n=0}^{\infty} \sum_{l=0}^{n} \sum_{m=-l}^{l} \Omega_{n,l,m}(f) Z_{n,l,m}, \tau_{(\theta,\phi)}(\sum_{n'=0}^{\infty} \sum_{l'=0}^{n'} \Omega_{n',l',0}(g) Z_{n',l',0}) \rangle \tag{13}$$

Using the properties of inner product, Eq. 13 can be rearranged as,

$$f * g(\theta, \phi) = \sum_{n=0}^{\infty} \sum_{l=0}^{n} \sum_{n'=0}^{\infty} \sum_{l'=0}^{n'} \sum_{m=-l}^{l} \Omega_{n,l,m}(f) \Omega_{n',l',0}(g) \langle Z_{n,l,m}, \tau_{(\theta,\phi)}(Z_{n',l',0}) \rangle \tag{14}$$

Using the rotational properties of Zernike polynomials, we obtain (see Appendix A for our full derivation),

$$f * g(\theta, \phi) = \frac{4\pi}{3} \sum_{n=0}^{\infty} \sum_{l=0}^{n} \sum_{m=-l}^{l} \Omega_{n,l,m}(f) \Omega_{n,l,0}(g) Y_{l,m}(\theta, \phi) \tag{15}$$

Since we can calculate $\Omega_{n,l,m}(f)$ and $\Omega_{n,l,0}(g)$ easily using Eq. 6, $f * g(\theta, \phi)$ can be found using a simple matrix multiplication. It is interesting to note that, since the convolution kernel does not translate, the convolution produces a polar shape, which can be further convolved–if needed–using the relationship $f * g(\theta, \phi) = \sqrt{\frac{4\pi}{2l+1}} \sum_{l} \sum_{m=-l}^{l} \hat{f}(l, m) \hat{g}(l, m) Y_{(l,m)}(\theta, \phi)$ where, $\hat{f}(l, m)$ and $\hat{g}(l, m)$ are the $(l, m)^{th}$ frequency components of $f$ and $g$ in spherical harmonics space. Next, we present a theorem to show the equivariance of volumetric convolution with respect to 3D rotation group.

### 4.3 EQUIVARIANCE TO 3D ROTATION GROUP

One key property of the proposed volumetric convolution is its equivariance to 3D rotation group. To demonstrate this, we present the following theorem.

**Theorem 1:** *Suppose $f, g : X \longrightarrow \mathbb{R}^3$ are square integrable complex functions defined in $\mathbb{B}^3$ so that $\langle f, f \rangle < \infty$ and $\langle g, g \rangle < \infty$. Also, let $\eta_{\alpha,\beta,\gamma}$ be a 3D rotation operator that can be decomposed into three Eular rotations $R_y(\alpha) R_z(\beta) R_y(\gamma)$ and $\tau_{\alpha,\beta}$ another rotation operator that can be decomposed into $R_y(\alpha) R_z(\beta)$. Suppose $\eta_{\alpha,\beta,\gamma}(g) = \tau_{\alpha,\beta}(g)$. Then, $\eta_{(\alpha,\beta,\gamma)}(f) * g(\theta, \phi) = \tau_{(\alpha,\beta)}(f * g)(\theta, \phi)$, where $*$ is the volumetric convolution operator.*

The proof to our theorem can be found in Appendix B. The intuition behind the theorem is that if a 3D rotation is applied to a function defined in $\mathbb{B}^3$ Hilbert space, the output feature map after volumetric convolution exhibits the same rotation. The output feature map however, is symmetric around north pole, hence the rotation can be uniquely defined in terms of azimuth and polar angles.

## 5 AXIAL SYMMETRY OF FUNCTIONS IN $\mathbb{B}^3$

In this section we present the following proposition to obtain the axial symmetry measure of a function in $\mathbb{B}^3$, around an arbitrary axis using 3D Zernike polynomials.

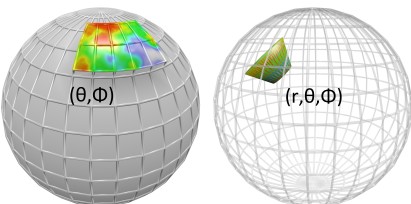

Figure 2: Kernel representations of spherical convolution (*left*) vs Volumetric convolution (*right*). In volumetric convolution, the shape is modeled and convolved in $\mathbb{B}^3$ which allows encoding non-polar 3D shapes with texture. In contrast, spherical convolution is performed in $\mathbb{S}^2$ that can handle only polar 3D shapes with uniform texture.

**Proposition:** *Suppose $g : X \longrightarrow \mathbb{R}^3$ is a square integrable complex function defined in $\mathbb{B}^3$ such that $\langle g, g \rangle < \infty$. Then, the power of projection of g in to $S = \{Z_i\}$ where S is the set of Zernike basis functions that are symmetric around an axis towards $(\alpha, \beta)$ direction is given by,*

$$||sym_{(\alpha,\beta)}|| = \sum_{n}\sum_{l=0}^{n} || \sum_{m=-l}^{l} \Omega_{n,l,m} Y_{m,l}(\alpha, \beta)||^2 \tag{16}$$

*where $\alpha$ and $\beta$ are azimuth and polar angles respectively.*

The proof to our proposition is given in Appendix C.

## 6 A CASE STUDY: 3D OBJECT RECOGNITION

### 6.1 3D OBJECTS AS FUNCTIONS IN $\mathbb{B}^3$

A 2D image is a function on Cartesian plane, where a unique value exists for any $(x, y)$ coordinate. Similarly, a polar 3D object can be expressed as a function on the surface of the sphere, where any direction vector $(\theta, \phi)$ has a unique value. To be precise, a 3D polar object has a boundary function in the form of $f : \mathbb{S}^2 \to [0, \infty]$.

Translation of the convolution kernel on $(x, y)$ plane in 2D case, extends to movements on the surface of the sphere in $\mathbb{S}^2$. If both the object and the kernel have polar shapes, this task can be tackled by projecting both the kernel and the object onto spherical harmonic functions (Appendix E). However, this technique suffers from two drawbacks. *1)* Since spherical harmonics are defined on the surface of the unit sphere, projection of a 3D shape function into spherical harmonics approximates the object to a polar shape, which can cause critical loss of information for non-polar 3D shapes. This is frequently the case in realistic scenarios. *2)* The integration happens over the surface of the sphere, which is unable to capture patterns across radius.

These limitations can be addressed by representing and convolving the shape function inside the unit ball ($\mathbb{B}^3$). Representing the object function inside $\mathbb{B}^3$ allows the function to keep its complex shape information without any deterioration since each point is mapped to unique coordinates $(r, \theta, \phi)$, where $r$ is the radial distance, $\theta$ and $\phi$ are azimuth and polar angles respectively. Additionally, it allows encoding of 2D texture information simultaneously. Figure 2 compares volumetric convolution and spherical convolution. Since we conduct experiments only on 3D objects with uniform surface values, in this work we use the following transformation to apply a simple surface function $f(\theta, \phi, r)$ to the 3D objects:

$$f(\theta, \phi, r) = \begin{cases} r, & \text{if surface exists at } (\theta, \phi, r) \\ 0, & \text{otherwise} \end{cases} \tag{17}$$

### 6.2 AN EXPERIMENTAL ARCHITECTURE

We implement an experimental architecture to demonstrate the usefulness of the proposed operations. While these operations can be used as building-tools to construct any deep network, we focus on three key factors while developing the presented experimental architecture: *1)* Shallowness: Volumetric convolution should be able to capture useful features compared to other methodologies with less number of layers. *2)* Modularity: The architecture should have a modular nature so that a fair comparison can be made between volumetric and spherical convolution. We use a capsule network after the convolution layer for this purpose. *3)* Flexibility: It should clearly exhibit the usefulness

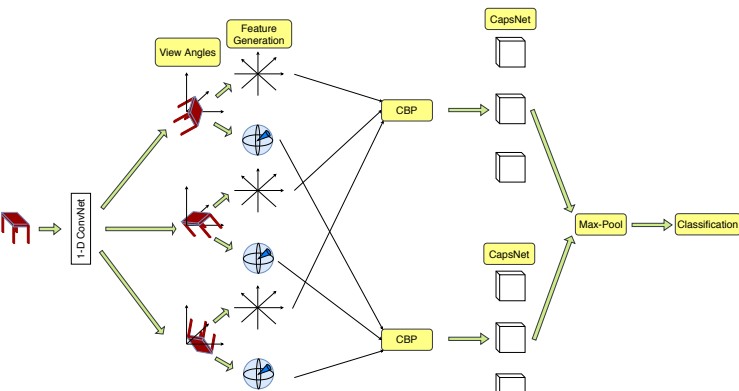

Figure 3: Experimental architecture: An object is first mapped to three view angles. For each angle, axial symmetry and volumetric convolution features are generated for $P^+$ and $P^-$. These two features are then separately combined using compact bilinear pooling. Finally, the features are fed to two individual capsule networks, and the decisions are max-pooled.

of axial symmetry features as a hand-crafted and fully differentiable layer. The motivation is to demonstrate one possible use case of axial symmetry measurements in 3D shape analysis.

The proposed architecture consists of four components. **First**, we obtain three view angles, and later generate features for each view angle separately. We optimize the view angles to capture complimentary shape details such that the total information content is maximized. For each viewing angle 'k', we obtain two point sets $P_k^+$ and $P_k^-$ consisting of tuples denoted as:

$$P_k^+ = \{(x_i, y_i, z_i) : y_i > 0\}, \text{ and } P_k^- = \{(x_i, y_i, z_i) : y_i < 0\}, \tag{18}$$

such that $y$ denotes the horizontal axis. **Second**, the six point sets are volumetrically convolved with kernels to capture local patterns of the object. The generated features for each point set are then combined using compact bilinear pooling. **Third**, we use axial symmetry measurements to generate additional features. The features that represent each point set are then combined using compact bilinear pooling. **Fourth**, we feed features from second and third components of the overall architecture to two independent capsule networks and combine the outputs at decision level to obtain the final prediction. The overall architecture of the proposed scheme is shown in Fig. 3.

### 6.3 OPTIMUM VIEW ANGLES

We use three view angles to generate features for better representation of the object. First, we translate the center of mass of the set of $(x, y, z)$ points to the origin. The goal of this step is to achieve a general translational invariance, which allows us to free the convolution operation from the burden of detecting translated local patterns. Subsequently, the point set is rearranged as an ordered set on $x$ and $z$ and a 1D convolution net is applied on $y$ values of the points. Here, the objective is to capture local variations of points along the $y$ axis, since later we analyze point sets $P^+$ and $P^-$ independently. The trained filters can be assumed to capture properties similar to $\partial^n y/\partial x^n$ and $\partial^n y/\partial z^n$, where $n$ is the order of derivative. The output of the 1D convolution net is rotation parameters represented by a $1 \times 9$ vector $\vec{r} = \{r_1, r_2, \cdots, r_9\}$. Then, we compute $R_1 = R_x(r_1)R_y(r_2)R_z(r_3)$, $R_2 = R_x(r_4)R_y(r_5)R_z(r_6)$ and $R_3 = R_x(r_7)R_y(r_8)R_z(r_9)$ where $R_1$, $R_2$ and $R_3$ are the rotations that map the points to three different view angles.

After mapping the original point set to three view angles, we extract the $P_k^+$ and $P_k^-$ point sets from each angle $k$ that gives us six point sets. These sets are then fed to the volumetric convolution layer to obtain feature maps for each point set. We then measure the symmetry around four equi-angular axes using Eq. 16, and concatenate these measurement values to form a feature vector for the same point sets.

### 6.4 FEATURE FUSION USING COMPACT BILINEAR POOLING

Compact bilinear pooling (CBP) provides a compact representation of the full bilinear representation, but has the same discriminative power. The key advantage of compact bilinear pooling is the significantly reduced dimensionality of the pooled feature vector.

We first concatenate the obtained volumetric convolution features of the three angles, for $P^+$ and $P^-$ separately to establish two feature vectors. These two features are then fused using compact bilinear

pooling (Gao et al., 2016). The same approach is used to combine the axial symmetry features. These fused vectors are fed to two independent capsule nets.

Furthermore, we experiment with several other feature fusion techniques and present results in Sec. 7.2.

### 6.5 CAPSULE NETWORK

Capsule Network (CapsNet) (Sabour et al., 2017) brings a new paradigm to deep learning by modeling input domain variations through vector based representations. CapsNets are inspired by so-called *inverse graphics*, i.e., the opposite operation of image rendering. Given a feature representation, CapsNets attempt to generate the corresponding geometrical representation. The motivation for using CapsNets in the network are twofold: *1)* CapsNet promotes a dynamic 'routing-by-agreement' approach where only the features that are in agreement with high-level detectors are routed forward. This property of CapsNets does not deteriorate extracted features and the final accuracy only depends on the richness of original shape features. It allows us to directly compare feature discriminability of spherical and volumetric convolution without any bias. For example, using multiple layers of volumetric or spherical convolution hampers a fair comparison since it can be argued that the optimum architecture may vary for two different operations. *2)* CapsNet provides an ideal mechanism for disentangling 3D shape features through pose and view equivariance while maintaining an intrinsic co-ordinate frame where mutual relationships between object parts are preserved.

Inspired by these intuitions, we employ two independent CapsNets in our network for volumetric convolution features and axial symmetry features. In this layer, we rearrange the input feature vectors as two sets of primary capsules—for each capsule net—and use the dynamic routing technique proposed by Sabour et al. (2017) to predict the classification results. The outputs are then combined using max-pooling, to obtain the final classification result. For volumetric convolution features, our architecture uses 1000 primary capsules with 10 dimensions each. For axial symmetry features, we use 2500 capsules, each with 10 dimensions. In both networks, decision layer consist of 12 dimensional capsules.

### 6.6 HYPERPARAMETERS

We use $n = 5$ to implement Eq. 15 and three iterations to calculate the Moore-Penrose pseudo inverse using Eq. 7. We use a decaying learning rate $lr = 0.1 \times 0.9^{\frac{g_{step}}{3000}}$, where $g_{step}$ is incremented by one per each iteration. For training, we use the Adam optimizer with $\beta_1 = 0.9, \beta_2 = 0.999, \epsilon = 1 \times 10^{-8}$ where parameters refer to the usual notation. All these values are chosen empirically. Since we have decomposed the theoretical derivations into sets of low-cost matrix multiplications, specifically aiming to reduce the computational complexity, the GPU implementation is highly efficient. For example, the model takes less than 15 minutes for an epoch during the training phase for ModelNet10, with a batchsize 2, on a single GTX 1080Ti GPU.

## 7 EXPERIMENTS

In this section, we discuss and evaluate the performance of the proposed approach. We first compare the accuracy of our model with relevant state-of-the-art work, and then present a thorough ablation study of our model, that highlights the importance of several architectural aspects. We use ModelNet10 and ModelNet40 datasets in our experiments. Next, we evaluate the robustness of our approach against loss of information and finally show that the proposed approach for computing 3D Zernike moments produce richer representations of 3D shapes, compared to the conventional approach.

### 7.1 COMPARISON WITH THE STATE-OF-THE-ART

Table 1 illustrates the performance comparison of our model with state-of-the-art. The model attains an overall accuracy of $92.17\%$ on ModelNet10 and $86.5\%$ accuracy on ModelNet40, which is on par with state-of-the-art. We do not compare with other recent work, such as Kanezaki et al. (2016); Qi et al. (2016); Sedaghat et al. (2016); Wu et al. (2016); Qi et al. (2016); Bai et al. (2016); Maturana & Scherer (2015) that show impressive performance on

| Method | Trainable layers | # Params | M10 | M40 |
|---|---|---|---|---|
| SO-Net (Li et al., 2018) | 11FC | 60M | 95.7% | 93.4% |
| Kd-Networks (Klokov & Lempitsky, 2017) | 15KD | - | 94.0% | 91.8% |
| VRN (Brock et al., 2016) | 45Conv | 90M | 93.11% | 90.8% |
| Pairwise (Johns et al., 2016) | 23Conv | 143M | 92.8% | 90.7% |
| MVCNN (Su et al., 2015) | 60Conv + 36FC | 200M | - | 90.1% |
| **Ours** | **3Conv + 2Caps** | **4.4M** | **92.17%** | **86.5%** |
| PointNet (Qi et al., 2017) | 2ST + 5Conv | 80M | - | 86.2% |
| ECC (Simonovsky & Komodakis, 2017) | 4Conv + 1FC | - | - | 83.2% |
| DeepPano (Shi et al., 2015) | 4Conv + 3FC | - | 85.45% | 77,63% |
| 3DShapeNets (Wu et al., 2015) | 4-3DConv + 2FC | 38M | 83.5% | 77% |
| PointNet (Garcia-Garcia et al., 2016) | 2Conv + 2FC | 80M | 77.6% | - |

Table 1: Comparison with state-of-the-art methods on ModelNet10 and ModelNet40 datasets (ranked according to performance). Ours achieve a competitive performance with the least network depth.

ModelNet10 and ModelNet40. These are not comparable with our proposed approach, as we propose a shallow, single model without any data augmentation, with a relatively low number of parameters. Furthermore, our model reports these results by using only a single volumetric convolution layer for learning features. Fig. 4 demonstrates effectiveness of our architecture by comparing accuracy against the number of trainable parameters in state-of-the-art models.

## 7.2 ABLATION STUDY

Table 3 depicts the performance comparison between several variants of our model. To highlight the effectiveness of the learned optimum view points, we replace the optimum view point layer with three fixed orthogonal view points. This modification causes an accuracy drop of 6.57%, emphasizing that the optimum view points indeed depends on the shape. Another interesting—perhaps the most important—aspect to study is the performance of the proposed volumetric convolution against spherical convolution. To this end, we replace the volumetric convolution layer of our model with spherical convolution and compare the results. It can be seen that our volumetric convolution scheme outperforms spherical convolution by a significant margin of 12.56%, indicating that volumetric convolution captures shape properties more effectively.

Furthermore, using mean-pooling instead of max-pooling, at the decision layer drops the accuracy to 87.27%. We also evaluate performance of using a single capsule net. In this scenario, we combine axial symmetry features with volumetric convolution features using compact bilinear pooling (CBP), and feed it a single capsule network. This variant achieves an overall accuracy of 86.53%, is a 5.64% reduction in accuracy compared to the model with two capsule networks.

Moreover, we compare the performance of two feature categories—volumetric convolution features and axial symmetry features—individually. Axial symmetry features alone are able to obtain an accuracy of 66.73%, while volumetric convolution features reach a significant 85.3% accuracy. On the contrary, spherical convolution attains an accuracy of 71.6%, which again highlights the effectiveness of volumetric convolution.

Then we compare between different choices that can be applied to the experimental architecture. We first replace the capsule network with a fully connected layer and achieve an accuracy of 87.3%. This is perhaps because capsules are superior to a simple fully connected layer in modeling view-

Table 2: Ablation study of the proposed architecture on ModelNet10 dataset

| Method | Accuracy |
|---|---|
| Final Architecture (FA) | 92.17% |
| FA + Orthogonal Rotation | 85.60% |
| FA - VolCNN + SphCNN | 79.53% |
| FA -MaxPool + MeanPool | 87.27% |
| FA + Feature Fusion (Axial + Conv) | 86.53% |
| Axial Symmetry Features | 66.73% |
| VolConv Features | 85.3% |
| SphConv Features | 71.6% |
| FA - CapsNet + FC layers | 87.3 % |
| FA - CBP + Feature concat | 90.7% |
| FA - CBP + MaxPool | 90.3% |
| FA - CBP + Average-pooling | 85.3 % |

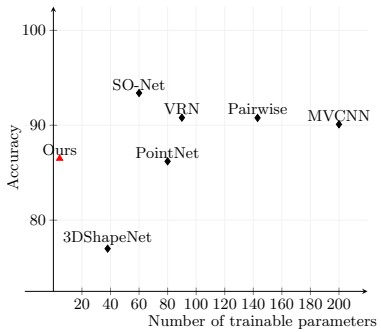

Figure 4: Accuracy vs number of trainable params (in millions) trend (ModelNet40)

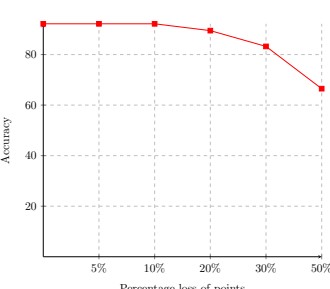
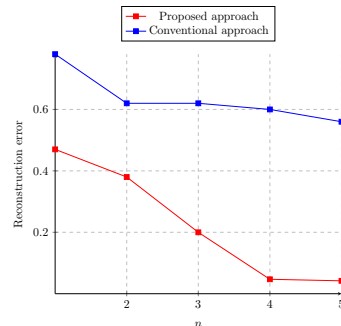

Figure 5: The robustness of the proposed model against missing data. The accuracy drop is less than 30% at a high data loss rate of 50%.

Figure 6: The mean reconstruction error Vs '$n$'. Our Zernike frequencies computation approach has far less error than the conventional approach.

point invariant representations. Then we try different substitutions for compact bilinear pooling and achieve 90.7%, 90.3% and 85.3% accuracies respectively for feature concatenation, max-pooling and average-pooling. This justifies the choice of compact bilinear pooling as a feature fusion tool. However, it should be noted that these choices may differ depending on the architecture.

### 7.3 ROBUSTNESS AGAINST INFORMATION LOSS

One critical requirement of a 3D object classification task is to be robust against various information loss. To demonstrate the effectiveness of our proposed features in this aspect, we randomly remove data points from the objects in validation set, and evaluate model performance. The results are illustrated in Fig. 5. The model shows no performance loss until 20% of the data is lost, and only gradually drops to an accuracy level of 66.5 at a 50% data loss, which implies strong robustness against random information loss.

### 7.4 EFFECTIVENESS OF THE PROPOSED METHOD FOR CALCULATING 3D ZERNIKE MOMENTS

In Sec. 4.1, we proposed an alternative method to calculate 3D Zernike moments (Eq. 5, 6), instead of the conventional approach (Eq. 3). We hypothesized that moments obtained using the former has a closer resemblance to the original shape, due to the impact of finite number of frequency terms. In this section, we demonstrate the validity of our hypothesis through experiments. To this end, we compute moments for the shapes in the validation set of ModelNet10 using both approaches, and compare the mean reconstruction error defined as: $\frac{1}{T} \sum_t^T \big\| f(t) - \sum_n \sum_l \sum_m \Omega_{n,l,m} Z_{n,l,m}(t) \big\|$, where $T$ is the total number of points and $t \in \mathbb{S}^3$. Fig. 6 shows the results. In both approaches, the mean reconstruction error decreases as $n$ increases. However, our approach shows a significantly low mean reconstruction error of $0.0467\%$ at $n = 5$ compared to the conventional approach, which has a mean reconstruction error of $0.56\%$ at same $n$. This result also justifies the utility of Zernike moments for modeling complex 3D shapes.

## 8 CONCLUSION

In this work, we derive a novel '*volumetric convolution*' using 3D Zernike polynomials, which can learn feature representations in $\mathbb{B}^3$. We develop the underlying theoretical foundations for volumetric convolution and demonstrate how it can be efficiently computed and implemented using low-cost matrix multiplications. Furthermore, we propose a novel, fully differentiable method to measure the axial symmetry of a function in $\mathbb{B}^3$ around an arbitrary axis, using 3D Zernike polynomials. Finally, using these operations as building tools, we propose an experimental architecture, that gives competitive results to state-of-the-art with a relatively shallow network, in 3D object recognition task. An immediate extension to this work would be to explore weight sharing along the radius of the sphere.

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

# Supplementary Material

# Volumetric Convolution: Automatic Representation Learning in Unit Ball

## A   CONVOLUTION WITHIN UNIT SPHERE USING 3D ZERNIKE POLYNOMIALS

**Theorem 1:**   *Suppose $f, g : X \longrightarrow \mathbb{R}^3$ are square integrable complex functions defined in $\mathbb{B}^3$ so that $\langle f, f \rangle < \infty$ and $\langle g, g \rangle < \infty$. Further, suppose $g$ is symmetric around north pole and $\tau(\alpha, \beta) = R_y(\alpha)R_z(\beta)$ where $R \in SO(3)$. Then,*

$$\int_0^1 \int_0^{2\pi} \int_0^{\pi} f(\theta, \phi, r), \tau_{(\alpha,\beta)}(g(\theta, \phi, r)) \sin \phi d\phi d\theta dr \equiv \frac{4\pi}{3} \sum_{n=0}^{\infty} \sum_{l=0}^{n} \sum_{m=-l}^{l} \Omega_{n,l,m}(f)\Omega_{n,l,0}(g)Y_{l,m}(\theta, \phi)$$

$$(19)$$

*where $\Omega_{n,l,m}(f), \Omega_{n,l,0}(g)$ and $Y_{l,m}(\theta, \phi)$ are $(n, l, m)^{th}$ 3D Zernike moment of $f$, $(n, l, 0)^{th}$ 3D Zernike moment of g, and spherical harmonics function respectively.*

**Proof:**   Since 3D Zernike polynomials are orthogonal and complete in $\mathbb{B}^3$, an arbitrary function $f(r, \theta, \phi)$ in $\mathbb{B}^3$ can be approximated using Zernike polynomials as follows.

$$f(\theta, \phi, r) = \sum_{n=0}^{\infty} \sum_{l=0}^{n} \sum_{m=-l}^{l} \Omega_{n,l,m}(f)Z_{n,l,m}(\theta, \phi, r) \tag{20}$$

where $\Omega_{n,l,m}(f)$ could be obtained using,

$$\Omega_{n,l,m}(f) = \int_0^1 \int_0^{2\pi} \int_0^{\pi} f(\theta, \phi, r)Z_{n,l,m}^{\dagger} r^2 sin\phi dr d\phi d\theta \tag{21}$$

where $^{\dagger}$ denotes the complex conjugate.

Leveraging this property (Eq. 20) of 3D Zernike polynomials Eq. 4 can be rewritten as,

$$f * g(\theta, \phi) = \langle \sum_{n=0}^{\infty} \sum_{l=0}^{n} \sum_{m=-l}^{l} \Omega_{n,l,m}(f)Z_{n,l,m}, \tau_{(\theta,\phi)}(\sum_{n'=0}^{\infty} \sum_{l'=0}^{n'} \sum_{m'=-l}^{l} \Omega_{n',l',m'}(g)Z_{n',l',m'}) \rangle \tag{22}$$

But since $g(\theta, \phi, r)$ is symmetric around $y$, the rotation around $y$ should not change the function. Which ensures,

$$g(r, \theta, \phi) = g(r, \theta - \alpha, \phi) \tag{23}$$

and hence,

$$\sum_{n'=0}^{\infty} \sum_{l'=0}^{n'} \sum_{m'=-l}^{l} \Omega_{n',l',m'}(g)R_{n',l'}(r)Y_{l',m'} = \sum_{n'=0}^{\infty} \sum_{l'=0}^{n'} \sum_{m'=-l}^{l} \Omega_{n',l',m'}(g)R_{n',l'}(r)Y_{l',m'}e^{-im'\alpha}$$

$$(24)$$

This is true, if and only if $m' = 0$. Therefore, a symmetric function around $y$, defined inside the unit sphere can be rewritten as,

$$\sum_{n'=0}^{\infty} \sum_{l'=0}^{n'} \Omega_{n',l',0}(g)Z_{n',l',0} \tag{25}$$

which simplifies Eq. 22 to,

$$f * g(\theta, \phi) = \langle \sum_{n=0}^{\infty} \sum_{l=0}^{n} \sum_{m=-l}^{l} \Omega_{n,l,m}(f) Z_{n,l,m}, \tau_{(\theta,\phi)}(\sum_{n'=0}^{\infty} \sum_{l'=0}^{n'} \Omega_{n',l',0}(g) Z_{n',l',0}) \rangle \tag{26}$$

Using the properties of inner product, Eq. 26 can be rearranged as,

$$f * g(\theta, \phi) = \sum_{n=0}^{\infty} \sum_{l=0}^{n} \sum_{n'=0}^{\infty} \sum_{l'=0}^{n'} \sum_{m=-l}^{l} \Omega_{n,l,m}(f) \Omega_{n',l',0}(g) \langle Z_{n,l,m}, \tau_{(\theta,\phi)}(Z_{n',l',0}) \rangle \tag{27}$$

Consider the term $\tau_{(\theta,\phi)}(Z_{n',l',0})$. Then,

$$\tau_{(\theta,\phi)}(Z_{n',l',0}) = \tau_{(\theta,\phi)}(R_{n',l'} Y_{l',0}) = R_{n',l'} \tau_{(\theta,\phi)}(Y_{l',0}) = R_{n',l'} \sum_{m''=-l'}^{l'} Y_{l',m''} D_{m'',0}^{l'}(\theta, \phi) \tag{28}$$

where $D_{m,m'}^{l}$ is the Wigner-D matrix. But we know that $D_{m'',0}^{l'}(\theta, \phi) = Y_{l',m''}(\theta, \phi)$. Then Eq. 27 becomes,

$$f * g(\theta, \phi) = \sum_{n=0}^{\infty} \sum_{l=0}^{n} \sum_{n'=0}^{\infty} \sum_{l'=0}^{n'} \sum_{m=-l}^{l} \Omega_{n,l,m}(f) \Omega_{n',l',0}(g) \sum_{m''=-l'}^{l'} Y_{l',m''}(\theta, \phi) \langle Z_{n,l,m}, Z_{n',l',m''} \rangle \tag{29}$$

$$f * g(\theta, \phi) = \frac{4\pi}{3} \sum_{n=0}^{\infty} \sum_{l=0}^{n} \sum_{m=-l}^{l} \Omega_{n,l,m}(f) \Omega_{n,l,0}(g) Y_{l,m}(\theta, \phi) \tag{30}$$

## B  EQUIVARIANCE OF VOLUMETRIC CONVOLUTION TO 3D ROTATION GROUP

**Theorem 1:** *Suppose $f, g : X \longrightarrow \mathbb{R}^3$ are square integrable complex functions defined in $\mathbb{B}^3$ so that $\langle f, f \rangle < \infty$ and $\langle g, g \rangle < \infty$. Also, let $\eta_{\alpha,\beta,\gamma}$ be a 3D rotation operator that can be decomposed into three Eular rotations $R_y(\alpha) R_z(\beta) R_y(\gamma)$ and $\tau_{\alpha,\beta}$ another rotation operator that can be decomposed into $R_y(\alpha) R_z(\beta)$. Suppose $\eta_{\alpha,\beta,\gamma}(g) = \tau_{\alpha,\beta}(g)$. Then,*

$$\eta_{(\alpha,\beta,\gamma)}(f) * g(\theta, \phi) = \tau_{(\alpha,\beta)}(f * g)(\theta, \phi) \tag{31}$$

*where $*$ is the volumetric convolution operator.*

**Proof:** Since $\eta_{(\alpha,\beta,\gamma)} \in SO(3)$, we know that $\eta_{(\alpha,\beta,\gamma)}(f(x)) = f(\eta_{(\alpha,\beta,\gamma)}^{-1}(x))$. Also we know that $\eta_{(\alpha,\beta,\gamma)} : \mathrm{R}^3 \to \mathrm{R}^3$ is an isometry.

Define,

$$\langle \eta_{(\alpha,\beta,\gamma)} f, \eta_{(\alpha,\beta,\gamma)} g \rangle = \int_{S^3} f(\eta_{(\alpha,\beta,\gamma)}^{-1}(x)) g(\eta_{(\alpha,\beta,\gamma)}^{-1}(x)) dx \tag{32}$$

Consider the Lebesgue measure $\lambda(S^3) = \int_{S^3} dx$. It can be proven that a lebesgue measure invariant under the isometries, which gives us $dx = d\eta_{(\alpha,\beta,\gamma)}(x) = d\eta_{(\alpha,\beta,\gamma)}^{-1}(x), \forall x \in S^3$. Therefore,

$$< \eta_{(\alpha,\beta,\gamma)} f, \eta_{(\alpha,\beta,\gamma)} g > = \int_{S^3} f(\eta_{(\alpha,\beta,\gamma)}^{-1}(x)) g(\eta_{(\alpha,\beta,\gamma)}^{-1}(x)) d(\eta_{(\alpha,\beta,\gamma)}^{-1} x) = < f, g > \tag{33}$$

Let $f(\theta, \phi, r)$ and $g(\theta, \phi, r)$ be the object function and kernel function (symmetric around north pole) respectively. Then volumetric convolution is defined as,

$$f * g(\theta, \phi) = < f, \tau_{(\theta,\phi)} g > \tag{34}$$

Applying the rotation $\eta_{(\alpha,\beta,\gamma)}$ to $f$, we get,

$$\eta_{(\alpha,\beta,\gamma)}(f) * g(\theta, \phi) = < \eta_{(\alpha,\beta,\gamma)}(f), \tau_{(\theta,\phi)} g > \tag{35}$$

By the result 33, we have,

$$\eta_{(\alpha,\beta,\gamma)}(f) * g(\theta, \phi) = < f, \eta_{(\alpha,\beta,\gamma)}^{-1}(\tau_{(\theta,\phi)} g) > \tag{36}$$

However, since $\eta_{\alpha,\beta,\gamma}(g) = \tau_{\alpha,\beta}(g)$ we get,

$$\eta_{(\alpha,\beta,\gamma)}(f) * g(\theta, \phi) = < f, \tau_{(\theta-\alpha,\phi-\beta,)} g > \tag{37}$$

We know that,

$$f * g(\theta, \phi) = < f, \tau_{(\theta,\phi)} g > = \sum_{n=0}^{\infty} \sum_{l=0}^{n} \sum_{m=-l}^{l} \Omega_{n,l,m}(f) \Omega_{n,l,0}(g) Y_{l,m}(\theta, \phi) \tag{38}$$

Then,

$$\eta_{(\alpha,\beta,\gamma)}(f) * g(\theta, \phi) = < f, \tau_{(\theta-\alpha,\phi-\beta)} g > = \sum_{n=0}^{\infty} \sum_{l=0}^{n} \sum_{m=-l}^{l} \Omega_{n,l,m}(f) \Omega_{n,l,0}(g) Y_{l,m}(\theta - \alpha, \phi - \beta) \tag{39}$$

$$\eta_{(\alpha,\beta,\gamma)}(f) * g(\theta, \phi) = (f * g)(\theta - \alpha, \phi - \beta) \tag{40}$$

$$\eta_{(\alpha,\beta,\gamma)}(f) * g(\theta, \phi) = \tau_{(\alpha,\beta)}(f * g) \tag{41}$$

Hence, we achieve equivariance over 3D rotations.

## C  AXIAL SYMMETRY MEASURE OF A FUNCTION IN $\mathbb{B}^3$ AROUND AN ARBITRARY AXIS.

**Proposition:** *Suppose $g : X \longrightarrow \mathbb{R}^3$ is a square integrable complex function defined in $\mathbb{B}^3$ such that $\langle g, g \rangle < \infty$. Then, the power of projection of $g$ in to $S = \{Z_i\}$ where $S$ is the set of Zernike basis functions that are symmetric around an axis towards $(\alpha, \beta)$ direction is given by,*

$$||sym_{(\alpha,\beta)}|| = \sum_{n}^{n} \sum_{l=0}^{l} || \sum_{m=-l}^{l} \Omega_{n,l,m} Y_{m,l}(\alpha, \beta) ||^2 \tag{42}$$

*where $\alpha$ and $\beta$ are azimuth and polar angles respectively.*

**Proof:** The subset of complex functions which are symmetric around north pole is $S = \{Z_{n,l,0}\}$. Therefore, projection of a function into $S$ gives,

$$sym_y(\theta, \phi) = \sum_{n}^{n} \sum_{l=0}^{n} \langle f, Z_{n,l,0} \rangle z_{n,l,0}(\theta, \phi) \tag{43}$$

To obtain the symmetry function around any axis which is defined by $(\alpha, \beta)$, we rotate the function by $(-\alpha, -\beta)$, project into $S$, and final compute the power of the projection.

$$sym_{(\alpha,\beta)}(\theta, \phi) = \sum_{n,l} \langle \tau_{(-\alpha,-\beta)}(f), Z_{n,l,0} \rangle z_{n,l,0}(\theta, \phi) \tag{44}$$

For any rotation operator $U$, and for any two points defined on a complex Hilbert space, $x$ and $y$,

$$\langle U(x), U(y) \rangle_H = \langle x, y \rangle_H \tag{45}$$

Applying this property to 44 gives,

$$sym_{(\alpha,\beta)}(\theta, \phi) = \sum_{n,l} \langle f, \tau_{(\alpha,\beta)}(Z_{n,l,0}) \rangle z_{n,l,0}(\theta, \phi) \tag{46}$$

Using Eq. 20 we get,

$$sym_{(\alpha,\beta)}(\theta, \phi) = \sum_{n}\sum_{l=0}^{n}\langle\sum_{n'}\sum_{l'=0}^{n'}\sum_{m'=-l'}^{l'} \Omega_{n'l'm'} Z_{n',l',m'}, \tau_{(\alpha,\beta)}(Z_{n,l,0}) \rangle z_{n,l,0}(\theta, \phi) \tag{47}$$

Using properties of inner product Eq. 47 further simplifies to,

$$sym_{(\alpha,\beta)}(\theta, \phi) = \sum_{n}\sum_{l=0}^{n}\sum_{n'}\sum_{l'=0}^{n'}\sum_{m'=-l'}^{l'} \Omega_{n'l'm'} \langle Z_{n',l',m'}, \tau_{(\alpha,\beta)}(Z_{n,l,0}) \rangle z_{n,l,0}(\theta, \phi) \tag{48}$$

Using the same derivation as in 28,

$$sym_{(\alpha,\beta)}(\theta, \phi) = \sum_{n}\sum_{l=0}^{n}\sum_{n'}\sum_{l'=0}^{n'}\sum_{m'=-l'}^{l'} \Omega_{n'l'm'} \sum_{m''=-l}^{l} Y_{l,m''}(\alpha, \beta) < Z_{n',l',m'}, Z_{n,l,m''} > z_{n,l,0}(\theta, \phi) \tag{49}$$

Since 3D Zernike Polynomials are orthogonal we get,

$$sym_{(\alpha,\beta)}(\theta, \phi) = \frac{4\pi}{3} \sum_{n}\sum_{l=0}^{n}\sum_{m=-l}^{l} \Omega_{n,l,m} Y_{m,l}(\alpha, \beta) z_{n,l,0}(\theta, \phi) \tag{50}$$

In signal theory the power of a function is taken as the integral of the squared function divided by the size of its domain. Following this we get,

$$||sym_{(\alpha,\beta)}|| = \langle (\sum_{n}\sum_{l=0}^{n}\sum_{m=-l}^{l} \Omega_{n,l,m} Y_{m,l}(\alpha, \beta)) z_{n,l,0}(\theta, \phi), (\sum_{n'}\sum_{l'=0}^{n'}\sum_{m'=-l'}^{l'} \Omega_{n',l',m'} Y_{m',l'}(\alpha, \beta) z_{n',l',0}(\theta, \phi))^{\dagger} \rangle \tag{51}$$

We drop the constants here since they do not depend on the frequency. Simplifying Eq. 51 gives,

$$||sym_{(\alpha,\beta)}|| = \sum_{n}\sum_{l=0}^{n}\sum_{m=-l}^{l}\sum_{m'=-l}^{l} \Omega_{n,l,m} Y_{m,l}(\alpha, \beta) \Omega_{n,l,m'} Y_{m',l}(\alpha, \beta) \tag{52}$$

which leads to,

$$||sym_{(\alpha,\beta)}|| = \sum_{n}\sum_{l=0}^{n}|| \sum_{m=-l}^{l} \Omega_{n,l,m} Y_{m,l}(\alpha, \beta)||^2 \tag{53}$$

## D  FUNCTION DEFINITIONS

### D.1  SPHERICAL HARMONICS

Spherical harmonics are a set of complete orthogonal functions, which are defined on $\mathbb{S}^2$.

$$Y_{l,m}(\theta, \phi) = (-1)^m \sqrt{\frac{2l+1}{4\pi} \frac{(l-m)!}{(l+m)!}} P_l^m(cos\phi)e^{im\theta} \tag{54}$$

where $l$ is an integer, $m$ is an integer, $|m| < l$, and $P_l^m(\cdot)$ is the associated Legendre function (see appendix D.2).

### D.2  ASSOCIATED LEGENDRE FUNCTION

Associated Legendre function $P_l^m(x)$ is defined as,

$$P_l^m(x) = (-1)^m \frac{(1-x^2)^{m/2}}{2^l l!} \frac{d^{l+m}}{dx^{l+m}} (x^2 - 1)^l \tag{55}$$

where $l$ is an integer, $m$ is an integer, $|m| < l$, and $x$ is a real number.

### D.3  ZERNIKE RADIAL POLYNOMIAL

$$R_{n,m}(r) = \sum_{k=0}^{(n-m)/2} \frac{(-1)^k (n-k)!}{k! \left((n+m)/2 - k\right)! \left((n-m)/2 - k\right)!} r^{n-2k} \tag{56}$$

## E  SPHERICAL CONVOLUTION ON $\mathbb{S}^2$

Let the $f$ and $g$ be the shape functions of the object and kernel respectively. Then $f$ and $g$ can be expressed as,

$$f(\theta, \phi) = \sum_l \sum_{m=-l}^l \hat{f}(l,m) Y_{l,m}(\theta, \phi) \quad \text{and,} \quad g(\theta, \phi) = \sum_l \sum_{m=-l}^l \hat{g}(l,m) Y_{l,m}(\theta, \phi) \tag{57}$$

where $Y_{l,m}$ is the $(l,m)^{th}$ spherical harmonics function and $\hat{f}(l,m)$ and $\hat{g}(l,m)$ are $(l,m)^{th}$ frequency components of $f$ and $g$ respectively. Then the frequency components of convolution $f * g$ can be easily calculated as,

$$\widehat{f * g}(l,m) = \sqrt{\frac{4\pi}{2l+1}} \hat{f}(l,m) \hat{g}(l,0)^\dagger \tag{58}$$

where $^\dagger$ denotes the complex conjugate.

## F  ROTATION PARAMETERS

Table 3: Average rotation parameter values across classes of ModelNet10. The values are reformatted to be positive angles between 0 and 360.

| Class | $r_1$ | $r_2$ | $r_3$ | $r_4$ | $r_5$ | $r_6$ | $r_7$ | $r_8$ | $r_9$ |
|---|---|---|---|---|---|---|---|---|---|
| Bathtub Bathtub | 319.2 | 100.5 | 57.8 | 185.2 | 223.4 | 98.3 | 350.6 | 167.4 | 14.2 |
| Bed | 264.3 | 196.3 | 103.7 | 208.5 | 186.2 | 194.4 | 267.9 | 246.3 | 81.2 |
| Chair | 198.6 | 91.2 | 243.7 | 47.4 | 161.2 | 87.9 | 240.5 | 47.3 | 203.4 |
| Desk | 88.4 | 80.2 | 130.9 | 206.6 | 86.5 | 112.8 | 291.7 | 233.2 | 351.4 |
| Dresser | 58.0 | 145.7 | 353.1 | 148.4 | 346.4 | 125.3 | 47.0 | 2.2 | 35.4 |
| Monitor | 218.9 | 279.0 | 58.1 | 10.4 | 30.3 | 331.4 | 90.7 | 285.6 | 346.1 |
| Night stand | 85.3 | 336.1 | 175.9 | 246.4 | 169.4 | 278.7 | 317.0 | 137.6 | 302.9 |
| Sofa | 306.1 | 86.9 | 109.2 | 311.1 | 22.5 | 321.4 | 96.9 | 47.0 | 76.2 |
| Table | 299.8 | 85.2 | 126.5 | 215.1 | 221.9 | 245.5 | 237.1 | 50.6 | 128.4 |
| Toilet | 277.0 | 325.3 | 215.5 | 255.6 | 192.2 | 19.8 | 278.4 | 193.4 | 348.2 |
| Average | 211.6 | 172.6 | 157.4 | 183.4 | 164.1 | 182.4 | 221.8 | 141.1 | 189.7 |

