# OpenReview forum: "Volumetric Convolution: Automatic Representation Learning in Unit Ball"
_ICLR.cc/2019/Conference_

### Official Review · AnonReviewer1 · 2018-11-03
**specific to unit-ball**

**Rating:** 5
**Confidence:** 3

**Review:**

Volumetric Convolution, Automatic Representation Learning in Unit Ball

This work proposes to tackle the challenging problem of learning on unit balls. The method uses volumetric convolutions based on the Zernike polynomial trick, which makes it convenient to use on convolution networks. Invariance to 3D rotation enables a transformation to a volumetric space, where convolutions could be used in a conventional process. Clarity of the methodology may benefit from a motivation discussed from a global perspective. The reader is currently facing heavy mathematical concepts fairly quickly with global rationale on the proposed choices, in particular, in the explanation of symmetry analysis. Clarity on the use of 2D and 3D features could also benefit from more details on what is exactly proposed. Results are shown on an object recognition task achieving performance comparable with the state-of-the-art.

Positive
+ Tackles the difficult problem of extending graph learning to arbitrary topologies, particularly on unit balls
+ The contributions are multifold -therotical framework for modeling volumetric convolutions over functions defined on unit-balls, -derivation of the formulation, to make it usable by neural nets, -measures of axial symmetry on unit-balls

Specific comments
- How to handle mixed topologies, for instance, with random presence of holes in the meshes
- Extension beyond unit balls?
- Fundamentaly, arbitrary genus-0 meshes are topologically equivalent to a sphere, however, there can be severe metric distorsion when transforming shapes to a sphere (e.g, transforming a banana to a sphere, the ends gets severely atrophied) - Does this pose a problem - how to handle these metric distorsion?
- Zernike polynomials are based on the spherical harmonics - could this be generalized to arbitrary graph harmonics? Beyond spherical shapes?

---

> ### Author Response · Authors · 2018-11-08
> **Thank you for your review**
>
> Thank you for your valuable review. Please find our answers below.
>
> - How to handle mixed topologies, for instance, with random presence of holes in the meshes
>
> Our technique is perfectly suitable for modeling 3D data with random presence of holes in the meshes. In fact, this is where our work stands out from other related techniques such as spherical convolution [1], which can only model polar shapes. Having random holes typically makes the shape non-polar. Since any 3D point cloud in B^3 can be parameterized using (\theta,\phi,r), the completeness property of Zernike polynomials (equation 2) ensures that they can reconstruct the shape with minimal loss. Moreover, [2] is some interesting work, which mention in their motivation that different genus shapes may be present for retrieval, for which they use 3D Zernike Polynomials.
>
> - Extension beyond unit balls?
>
> Since any 3D point cloud can be modeled as a function in B^3, we do not see this as a valid requirement. Can you please be kind enough to clarify your question?
>
> - Fundamentally, arbitrary genus-0 meshes are topologically equivalent to a sphere, however, there can be severe metric distortion when transforming shapes to a sphere (e.g, transforming a banana to a sphere, the ends gets severely atrophied) - Does this pose a problem - how to handle these metric distortion?
>
> Theoretically, as we mentioned under point 1, an arbitrary 3D point cloud can be efficiently modeled as a function in B^3 (as opposed to spherical projection), without loss or distortion of information.
>
> - Zernike polynomials are based on the spherical harmonics - could this be generalized to arbitrary graph harmonics? Beyond spherical shapes?
>
> Spherical harmonics exhibit a "form invariance" to rotation, which is important for achieving rotation equivariance (see equation 28). Also, our method is not limited to spherical shapes, and can model any 3D shape (as we have experimentally and theoretically shown in our work).
>
> [1] - Taco S. Cohen, Mario Geiger, Jonas Köhler, Max Welling, Spherical CNNs. International Conference on Learning Representations (ICLR), 2018. (https://arxiv.org/abs/1801.10130)
>
> [2] Novotni, Marcin, and Reinhard Klein. "Shape retrieval using 3D Zernike descriptors." Computer-Aided Design 36.11 (2004): 1047-1062.( http://citeseerx.ist.psu.edu/viewdoc/download?doi=10.1.1.71.8238&rep=rep1&type=pdf )
>
> Please let us know if you have any further questions or concerns. Thanks a lot for your valuable time.

---

### Official Review · AnonReviewer3 · 2018-11-05
**Closely related to the recent spherical CNN and SE(n) equivariant network papers, but mathematically less clean**

**Rating:** 5
**Confidence:** 5

**Review:**

There is a great amount of interest in extending the notion of equivariance in neural networks from
just translations to other groups. In particular, in the past year a sequence of papers have appeared
starting with (Cohen, Geiger et al.) on "spherical CNNs" that achieve equivariance to rotations for images
painted on the surface of the unit sphere.

The present paper extends these ideas to volumetric data in the unit ball (rather than just the sphere)
by the use of Zernike polynomials. Since Zernike polynomials can be expressed as the product of spherical
harmonics with a radial function, this is essentially the same as adding a radial component to a spherical
CNN.

The main result of the paper appears to be Theorem 1, which shows that what the authors define as
volumetric convolution is equivariant to rotations. This is split across Sections 4.2 and 4.3.. However,
apart from the radial component, this result is bascally the same as the SO3-equivariance of spherical CNNs,
as discussed in three very recent spherical CNN papers: (Cohen, Geiger et al.) (Esteves Allen-Blanchette et al)
and (Kondor, Lin and Trivedi). However, the somewhat more abstract, representation theoretic approach
of some of these works allows a more compact derivation than the one in the present paper.

The authors also fail to cite recent work on SE(2) and SE(3) equivariant neural networks. SE(3) comprises
all rotations and translations of R^3, so the latter, in particular, encapuslates SO(3) equivariance as
a special case. In particular, part of the construction in (Weiler, Hamprecht and Storath, CVPR 2018)
is to add  Gaussian radial functions to SO(2) equivariant filters, which is just the 2D analog of
what is happening in the present paper. Then in (Weiler, Geiger, et al., 2018) the same is done in 3D,
except of course they go further by also adding translation equivariance. Admittedly, these papers are
very new, so the authors might not have known about them.

I also find some of the mathematical details a little puzzling:

1. As explained in the spherical CNN papers, taking the cross correlation of two functions on the sphere
(by extension, in the unit ball) naturally results in a function that lives on the rotation group SO(3), i.e.
the cross-correlation (or convolution) is parametrized by three Euler angles. I don't understand why the
authors restrict themselves to considering just two angles, forcing their filters to be polar, as
derived in Section 5. This seems like an artificial restriction that will limit the power of their approach.

2. The paper mentions that Zernike polynomials are "orthogonal and complete in B^3". I think what they mean
is that they are an ortogonal and complete basis for an appropriate space of functions on B^3, and that
space of functions is L_2(B^3). However, this is still not enough. For (3) to hold, one also needs the
basis to be orthonormal. Please be more precise.

3. In the same vein, at one point in the proof, the authors mention that "rotations are unitary operators
in a Hilbert space". This is not true of Hilbert spaces in general. It requires the above orthonormality etc..

4. Exactly as a consequence of orthonorlamity, Equation 5 is essentially just a generalized Fourier transform
on B^3, hence, in principle, it can be inverted analytically. I understand that the fact that the input
image is rasterized complicates things and in a practical implementation it might be expedient to invert (5)
by using the pseudo-inverse. However, this is a one-time operations and is really just a hack. It seems strange
that the authors use a special iterative method just to invert a close to unitary matrix.

The mathematical shortcomings of the paper could be compensated by amazingly good experimental results.
The actual results are good, but still not best-in-class, possibly because at the end of the day the network
is still only rotationally equivariant and does not take into account translations.

The spherical CNN and SE(n) equivariance papers generally apply the group equivariant operations consistently
across multiple layers. In contrast, the present paper only applies it in the first layer, and then uses a
combination of tricks like multiple viewpoints and bilinear pooling to boost performance. Unfortunately, the
benefits of this additional conceptual complexity are not entirely borne out in the experimental results.

---

> ### Author Response · Authors · 2018-11-07
> **Thank you for your review**
>
> Thanks for your valuable comments. Please find our responses below.
>
> We thank you for mentioning the closely related papers. However, we would like to point out that the proposed integration across radius provides critical advantages compared to mentioned work which work on S^2 (we have stated this in the paper): 1) Since spherical harmonics are defined on the surface of the unit sphere, projection of a 3D shape function into spherical harmonics approximates the object to a polar shape, which can cause critical loss of information for non-polar 3D shapes. This is a common case in realistic scenarios. 2) In spherical convolution, the integration happens over the surface of the sphere, which cannot capture patterns across radius.
>
> Also, we would like to pointout that to the best of our knowledge, our work is the first approach to perform volumetric convolution on B^3 that can simultaneously model 2D (appearance) and 3D (shape) features.
>
> Furthermore, we have clearly demonstrated through experiments that volumetric convolution's integration across radius provides superior results over spherical convolution (see Table 2). Therefore, we would like to reiterate that this work has a clear novelty and advantage over competing methods.
>
> We also briefly studied the mentioned Weiler, Geiger, et al., 2018, and would like to point out that their work focuses in modeling 3D data as dense vector fields in 3D Euclidean space. In contrast (as you also have mentioned), we model the 3D point clouds in B^3, and is a fundamentally different approach for tackling a similar problem. We would cite their work accordingly. Thank you for mentioning this paper.
>
>
> 1. The use of two angles for rotation of the kernel originates from forcing the kernels to be symmetric around north pole. This property is essential for the clean derivation of our Theorem 1.
>
> For example, without the symmetry, stepping from Eq. 22 to 26 is not possible. Eq. 26 is vital because it contains Z_{n',l',0}.  It is only because of Z_{n',l',0} we get D^l'_{m'',0} in Eq. 28, which is critical for the final result in Eq. 30. We would like to provide more clarification on this, if our explanation is not clear enough.
>
>
> 2,3- Our derivations do not require orthonormality to hold. The requirement is the rotation operator to preserve the inner product of square integral functions in B^3 as presented in Eq. 45. We prove that this requirement is satisfied in Eq. 32-33. We agree with your comment regarding the tern "unitary". We would remove this term in the revised paper.
>
> 4. First, we avoid using SVD or any other non-differentiable analytical method to calculate the inverse of matrix to preserve the differentiability of the pipeline. Therefore, volumetric convolution layer can be implemented as a fully differentiable layer, which can be integrated into any deep architecture. We specifically mention this fact in the paper.
>
> Second, this is not a one time operation if multiple layers exist. As it is clear from Eq. 8, the result from convolution gives the response in spatial domain (this is different from spherical convolution, where the convolution in spatial domain corresponds to the multiplication in spectral domain, which allows one time conversion to spectral domain across multiple layers). Therefore, it is vital to reduce the computational complexity as much as possible.
>
> 5. Our main focus in this paper is to present the theory of volumetric convolution, implement it as a fully differentiable layer, and demonstrate it's superiority of calculating richer features compared to methods such as spherical convolution. The reason for not using multiple layers is mentioned in the paper as follows.
>
> "CapsNet promotes a dynamic ‘routing-by-agreement’ approach where only the features that are in agreement with high-level detectors are routed forward. This property of CapsNets does not deteriorate extracted features and the final accuracy only depends on the richness of original shape features. It allows us to directly compare feature discriminability of spherical and volumetric convolution without any bias.  For example, using multiple layers of volumetric or spherical convolution hampers a fair comparison since it can be argued that the optimum architecture may vary for two different operations."
>
> Also, we experimentally demonstrate the superiority of the proposed volumetric convolution over spherical convolution in Table 2. Investigating the optimum architecture with multiple layers of volumetric convolution is out of the scope of this work.
>
> Please let us know if you have any further questions or concerns. Thanks a lot for your valuable time.

---

> > ### Comment · AnonReviewer3 · 2018-11-12
> > **Thank you for your response.**
> >
> > Your response does not alleviate my concerns, it seems like some of my points have been misunderstood.
> >
> > 0.  I understand that the proposed method is more powerful than Spherical CNNs. My assertion in the review was that given the Spherical CNN architecture, adding a radial component is relatively trivial, and this is essentially what is happening in this paper.
> >
> > Moreover, [Weiler, et al., 2018] have already solved the more general problem of SE(3) equivariance. Your objection of them modeling dense vector fields rather than point clouds is not a fundamental difference at the conceptual level.
> >
> > 1. If you considered kernels that are not polar (symmetric around the N pole), then the algorithm would likely be more powerful. In general, the convolution of two functions on S^2 or on B^3 wrt to the action of the rotation group SO(3) is a function on SO(3) rather than on S^2. I believe this is explained in [Cohen, Gieger et al, 2018], and several other places.
> >
> > 4. I understand that transforming to the Zernike basis needs to be done in every layer. However, the transform itself is fixed, it does need to be learned. This is the sense in which I say that computing the transform matrix is a "one-time operation". It could be done analytically. It is just a constant matrix that the activations of the previous layer are multiplied by. Multiplying by a constant matrix is easy to implement in any framework.

---

> > > ### Author Response · Authors · 2018-11-14
> > > **Thank you for insightful comments**
> > >
> > > We thank you for the insightful and valuable reply. Please find our responses below.
> > >
> > > Q0 - Radial component  and Weiler, et al., 2018
> > >
> > > We agree with your point that the addition of the radial component is a straightforward extension. However, the radial component essentially moves the functions from S^2 to B^3, which provides several fundamental advantages (as mentioned in the paper). We believe that these advantages cannot be undermined and therefore our work provides a novel contribution. We would also like to mention that our secondary result, the axial symmetry measurement, (to the best of our knowledge)  is the first to present a fully differentiable technique to measure the axial symmetry of functions in B^3.
> > >
> > > We completely agree that Weiler, et al., 2018 has proposed an effective method to achieve SE(3) equivariance. Thank you for bringing our attention to this very recent paper which we were not aware of. We have cited this paper in the revised version.
> > >
> > > However, we think that modeling functions in B^3 opens up a new path to achieve the roto-translational equivariance. For example, although we only achieve equivariance over rotation in this work due to inherent limitations of the radial component of the Zernike polynomials, it is possible to achieve translation equivariance with a suitable radial component. Since to the best of our knowledge, the presented work is the first to present the *theory* of volumetric convolution in B^3 using orthogonal basis functions (with angular and radial components), we believe our work will lead to future work (targeting rotational and translation equivariance) which follows this path. We would also like to mention that this cannot be achieved using convolutions in S^2.
> > >
> > > Q1 - Cross-correlation as a function of SO(3)
> > >
> > > We understand that getting cross-correlation between two functions on S^2 as a function of SO(3) is likely to be more powerful. However, implementing cross-correlation between two functions in B^3 as a function of SO(3) is not as straightforward to be reduced to a set of matrix/tensor operations, as in the case of S^2.
> > >
> > > More precisely, cross-correlation between two functions on S^2 can be written as follows.
> > >
> > > \sum_{l,m,m'}\hat{f}_{l,m}\hat{g}_{l,m'}D^l_{m,m'}
> > >
> > > The Fourier transform of the cross correlation then just becomes the outer product of \hat{f}_{l,m} and hat{g}_{m,m'}
> > >
> > > However, in the case of B^3, the equation becomes,
> > >
> > > \sum_{n}\sum_{l,m,m'}\hat{f}_{n,l,m}\hat{g}_{n,l,m'}D^l_{m,m'}
> > >
> > >
> > > Where the outer product is no longer equal to the Fourier transform of the cross correlation. Moreover, the condition that n-l should be even, makes it more difficult to reduce this operation to a matrix/tensor operation while preserving the differentiability.
> > >
> > > Q4 - Analytical inversion
> > >
> > > An analytical method could have been used, under the condition, that the inputs to the volumetric layer do not depend on any learnable function. However, our work is not limited to such networks and demonstrates this by having a 1-D ConvNet below the volumetric convolution layer. The transformation matrix depends on the rotated points in the point cloud by 1-D ConvNet. (We choose rotated points in the point cloud to determine the X in equation 6). Therefore, it has to be determined dynamically during the forward pass. Hence, as we needed to preserve the differentiability in the Volumetric Convolution layer, we avoided integrating any non-differentiable analytical method to compute the inverse of X. This allows problem-free error propagation across the volumetric convolution layer to the lower layers during the training phase (in our case, the 1-D ConvNet). Therefore, having the iterative method enables our volumetric convolution layer to be integrated to any system, which is trained via back-prop.
> > >
> > > Please let us know if you have further concerns/questions. Thanks a lot for your time.

---

### Official Review · AnonReviewer4 · 2018-11-08
**Benefit for volumetric data not clear**

**Rating:** 6
**Confidence:** 2

**Review:**

The work concerns convolution in the unit sphere. It differentiates itself from previously mentioned work by working in the volume space and not the surface space. While I can't say I understand all of the implications of the work,  I was left with several questions. Many of these questions are in regards to claims made by the authors whose answer or reference was not made clear.

- It was not made clear why there is a benefit to convolving the object in the unit sphere vs the unit cube, especially given that the work was not able to perform better than other work that was based on the unit sphere. This point was the stated problem of the paper. Although it was mentioned that the unit sphere preserves all of the points of the object, it isn't clear if the transformation causes any deformations of the object. Furthermore, the fixing of one axis seems to be a way to hack around problems of increased dimensionality, but there was no justification given.

- How does the number of trainable layers help to differentiate resource usage. Wouldn't a better measure be number of parameters? The authors make that claim that shallowness is a virtue, but there is little discussion as to the size of each layer in comparable terms.

- Why was no "ablation" or "accuracy vs trained layers" data shown for the Modelnet40 dataset? I would think that would be stronger evidence than for the Modelnet10 data.

- Why wasn't the 1d conv net used for creating the viewing angles included in the size of the architecture? Was there a verification as to what the filters from this network were actually giving? The authors mention how we should interpret them, but not enough information about the structure of the network is given to satisfy this question.

- I would have liked to see a description of the types of features that are found by these networks.

- The authors say they are only going to show experiments on one possible use case, but then make claims for other use cases. I am referencing that since the texture data in the datasets used is constant, there was no need to model the texture data. There is no experimental evidence to show this is the case, however.

Overall, I think the paper would have been stronger if it had more experiments.

---

> ### Author Response · Authors · 2018-11-11
> **Thank you for your review**
>
> Thank you for the insightful comments. Please find our answers below.
>
> Q1 - The representation of a 3D function defined inside a cube is less convenient for extracting rotation invariants, compared to a sphere and a ball. For example Cantarkis [1] showed that the a set of complete basis functions defined inside a unit ball can be formed as Z = R(x)Y(\theta,\phi). Which means the linear and angular representations can be decomposed and analyzed efficiently. Achieving rotational equivariance depends only on the angular part while the translational equivariance depends only in the linear part. While we do not achieve translational equivariance in this work due to inherent limitations of the linear component of the Zernike polynomials, it might be possible to achieve this task with a suitable linear component.
>
> We have shown in table 2 that our volumetric convolution layer produce richer features compared to closely related spherical convolution methods, which operates on the surface of the sphere. In fact, the volumetric convolution layer achieves a significant 13.7% gain over spherical convolution layer. Convolving in B^3 provides critical advantages compared to other work which work on S^2: 1) Since spherical harmonics are defined on the surface of the unit sphere, projection of a 3D shape function into spherical harmonics approximates the object to a polar shape, which can cause critical loss of information for non-polar 3D shapes. This is a common case in realistic scenarios. 2) In spherical convolution, the integration happens over the surface of the sphere, which cannot capture patterns across radius.
>
> The deformations depends on the order of moments used. For example, the transformation can be considered as a low pass filter, where higher the order of moments, sharper the subtle variations and details of the original object would be represented in the spectral domain. We have derived a novel technique to reduce the distance between the perfect shape representation and practical representation (please see section 4.1). We have practically evaluated and presented the effectiveness of our technique by calculating the reconstruction error in Figure 5 and Figure 6, where we achieve a very low reconstruction error of 0.0467%, which means the deformation can be neglected.
>
> Q2 - We agree that number of parameters would be a valuable comparison across architectures. We would add this discussion after investigating the number of parameters in competing methods. Thank you.
>
> Q3 - The "accuracy vs trained layers" data is illustrated in table 1. We have added a graphical illustration only for ModelNet10. We will replace it with ModelNet40 data.
>
> Q4 - We accept it is a mistake by us to not to include the 1-D convnet to the size of the architecture. We have corrected it in the revised version. We  did not interpret the outputs of the 1-D convnet as it was not the main focus of the paper.
>
> Q5 - Analyzing functions in B^3 is vital not only for 3D object analysis but in other areas such as function analysis and quantum optics/wavelet analysis ([2], [3]). Therefore, we did not limit our discussion to 3D object recognition, although we experiment only on one use case. Please consider that this paper is mainly a theoretical paper, where we do the practical implementation as a fully differentiable technique, so it can be integrated into any system that is trained via back-propagation.
>
> Since the lack of 3D object datasets which have texture, we add a simple surface function to the available datasets (equation 17). In cases such as wavelet analysis, the ability to model texture is highly important as the intensity if the points play a key role. Note that this task cannot be achieved by existing techniques in literature which operates on S^2.
>
>
> [1] - Canterakis, N. "3D Zernike moments and Zernike affine invariants for 3D image analysis and recognition." In 11th Scandinavian Conf. on Image Analysis. 1999. (http://citeseerx.ist.psu.edu/viewdoc/summary?doi=10.1.1.16.3441)
>
> [2] - Janssen, Augustus JEM. "Generalized 3D Zernike functions for analytic construction of band-limited line-detecting wavelets." arXiv preprint arXiv:1510.04837 (2015). (https://arxiv.org/abs/1510.04837)
> [3] - Torre, A. "Generalized Zernike or disc polynomials: an application in quantum optics." Journal of Computational and Applied Mathematics 222.2 (2008): 622-644. (https://www.sciencedirect.com/science/article/pii/S0377042707006450)
>
> Please let us know if you have further concerns that we can clarify. Thank you for your valuable time.

---

> ### Author Response · Authors · 2018-11-18
> **Revised version uploaded.**
>
> Thank you for your valuable suggestions which helped us to improve our manuscript. In the revised version, we have added the following content.
>
> 1. We have added the comparison of number of trainable parameters across state-of-the-art architectures. The Table 1 contains this new information. This comparison clearly shows the efficiency of our architecture, as we have only 4.4M trainable parameters (~9x less) compared to 3D shape net (architecture with the second lowest number of parameters), which has 38M parameters.
>
> 2. We have updated Figure 4 with ModelNet40 data. Also, we changed x axis from trainable layers to trainable parameters.
>
> 3. 1-D ConvNet is added to the number of trainable layers (Table 1)
>
> Thank you for these suggestions. Please let us know if you have further questions/concerns.

---

> ### Author Response · Authors · 2018-11-23
> **Outputs of 1-D ConvNet**
>
> As per the point regarding the outputs of the 1-D ConvNet, we conducted an experiment to obtain the rotation parameters across the classes in ModelNet10.
>
> The average values of the rotation parameters given by the 1-D convnet (across ModelNet10 classes) are shown below. Please note that we have reformatted the parameter values to be a positive angle between 0-360.
>
>
> -----------------------------------------------------------------------------------------------------------
> | Class       | r1       | r2      | r3     | r4       | r5       | r6     | r7       | r8      | r9     |
> -----------------------------------------------------------------------------------------------------------
> | Bathtub  | 319.2 | 100.5 | 57.8  | 185.2 | 223.4 | 98.3  | 350.6 | 167.4 | 14.2  |
> -----------------------------------------------------------------------------------------------------------
> | Bed         | 264.3 | 196.3 | 103.7| 208.5 | 186.2 | 194.4| 267.9 | 246.3 | 81.2  |
> -----------------------------------------------------------------------------------------------------------
> | Chair       | 198.6 | 91.2  | 243.7 | 47.4  | 161.2  | 87.9  | 240.5 | 47.3  | 203.4 |
> -----------------------------------------------------------------------------------------------------------
> | Desk       | 88.4  | 80.2   | 130.9 | 206.6| 86.5    | 112.8| 291.7 | 233.2| 351.4  |
> -----------------------------------------------------------------------------------------------------------
> | Dresser  | 58.0  | 145.7 | 353.1 | 148.4 | 346.4 | 125.3 | 47.0  | 2.2    | 35.4    |
> -----------------------------------------------------------------------------------------------------------
> | Monitor | 218.9 | 279.0| 58.1   | 10.4   | 30.3   | 331.4 | 90.7  | 285.6 | 346.1 |
> -----------------------------------------------------------------------------------------------------------
> |Night std| 85.3  | 336.1 | 175.9 | 246.4 | 169.4 | 278.7 | 317.0| 137.6 | 302.9 |
> -----------------------------------------------------------------------------------------------------------
> | Sofa        | 306.1 | 86.9 | 109.2 | 311.1 | 22.5   | 321.4 | 96.9  | 47.0   | 76.2   |
> -----------------------------------------------------------------------------------------------------------
> | Table       | 299.8 | 85.2 | 126.5 | 215.1 | 221.9 | 245.5 | 237.1| 50.6  | 128.4 |
> -----------------------------------------------------------------------------------------------------------
> | Toilet      | 277.0 | 325.3| 215.5 | 255.6 | 192.2 | 19.8  | 278.4 | 193.4| 348.2 |
> -----------------------------------------------------------------------------------------------------------
> | Average | 211.6 | 172.6| 157.4 | 183.4 | 164.1 | 182.4| 221.8 | 141.1| 189.7 |
> -----------------------------------------------------------------------------------------------------------
>
> We have included this information in Appendix F. Thank you for your suggestions for improving our manuscript. We would be happy to address any more concerns you have regrding our work.

---

### Meta-Review · Area_Chair1 · 2018-12-12
**good paper, but some improvement possible**

**Confidence:** 3
**Recommendation:** Reject

**Metareview:**

Using volumetric convolutions, this paper focuses on learning in (rather than on) the unit sphere.

The novelty of the approach is debatable, and the mathematical analysis not strong enough to merit that.  In combination with good but not outstanding results, interest of the research community is doubted.  An extended experimental analysis of the method would greatly improve the paper.